# Chromatin mapping and single-cell immune profiling define the temporal dynamics of ibrutinib response in CLL

André F. Rendeiro [1,7], Thomas Krausgruber [1,7], Nikolaus Fortelny[1], Fangwen Zhao[1,2], Thomas Penz[1], Matthias Farlik[1], Linda C. Schuster[1], Amelie Nemc[1], Szabolcs Tasnády[3], Marienn Réti[3], Zoltán Mátrai[3], Donát Alpár[1,4,8], Csaba Bödör [4,8], Christian Schmidl[1,6,8] & Christoph Bock [1,2,5,8]*

The Bruton tyrosine kinase (BTK) inhibitor ibrutinib provides effective treatment for patients with chronic lymphocytic leukemia (CLL), despite extensive heterogeneity in this disease. To define the underlining regulatory dynamics, we analyze high-resolution time courses of ibrutinib treatment in patients with CLL, combining immune-phenotyping, single-cell transcriptome profiling, and chromatin mapping. We identify a consistent regulatory program starting with a sharp decrease of NF-κB binding in CLL cells, which is followed by reduced activity of lineage-defining transcription factors, erosion of CLL cell identity, and acquisition of a quiescence-like gene signature. We observe patient-to-patient variation in the speed of execution of this program, which we exploit to predict patient-specific dynamics in the response to ibrutinib based on the pre-treatment patient samples. In aggregate, our study describes time-dependent cellular, molecular, and regulatory effects for therapeutic inhibition of B cell receptor signaling in CLL, and it establishes a broadly applicable method for epigenome/transcriptome-based treatment monitoring.

[1]CeMM Research Center for Molecular Medicine of the Austrian Academy of Sciences, Vienna, Austria. [2] Ludwig Boltzmann Institute for Rare and Undiagnosed Diseases, Vienna, Austria. [3] Department of Haematology and Stem Cell Transplantation, Central Hospital of Southern Pest, National Institute of Hematology and Infectious Diseases, Budapest, Hungary. [4] MTA-SE Lendület Molecular Oncohematology Research Group, 1st Department of Pathology and Experimental Cancer Research, Semmelweis University, Budapest, Hungary. [5] Department of Laboratory Medicine, Medical University of Vienna, Vienna, Austria. [6]Present address: Regensburg Center for Interventional Immunology (RCI), Regensburg, Germany. [7]These authors contributed equally: André F. Rendeiro, Thomas Krausgruber. [8]These authors jointly directed this work: Donát Alpár, Csaba Bödör, Christian Schmidl, Christoph Bock. *email: cbock@cemm.oeaw.ac.at

Chronic lymphocytic leukemia (CLL) is among the most frequent blood cancers[1]. It is a disease characterized by extensive clonal proliferation and accumulation of malignant B lymphocytes in the blood, bone marrow, spleen, and lymph nodes. On a cellular level, this process is driven by constitutively activated B-cell receptor (BCR) signaling, which can be caused by erroneous (auto)antigen recognition and/or cell-autonomous mechanisms[2].

CLL shows remarkable clinical heterogeneity, with some patients pursuing an indolent course, while others progress rapidly and require early treatment. Extensive heterogeneity exists also at the genetic, epigenetic, and transcriptional level. Recent research has identified genetically defined CLL subtypes[3–6] and patient-specific transcriptional programs[7–9]. Moreover, characteristic DNA methylation patterns appear to reflect differences in the CLL's cell-of-origin[10–12], and chromatin profiles predict the BCR immunoglobulin heavy-chain variable (IGHV) gene mutation status[13].

Despite widespread clinical and molecular heterogeneity, therapeutic inhibition of BCR signaling using the Bruton tyrosine kinase (BTK) inhibitor ibrutinib[14] has remarkable efficacy in essentially all patients with CLL. Most notably, ibrutinib treatment achieves high clinical response rates even in patients carrying high-risk genetic markers predictive of fast disease progression, such as TP53 aberrations[15–18]. Due to its excellent clinical efficacy and usually tolerable side effects, ibrutinib treatment is becoming the standard of care for most patients with CLL that require treatment.

Successful ibrutinib therapy often causes an initial increase of CLL cells in peripheral blood that can take months to resolve[19,20]. This observation has been explained by the drug's effect on cell–cell contacts[21,22], which triggers relocation of CLL cells from their protective microenvironment to the peripheral blood. As the result of this ibrutinib-induced lymphocytosis, the correlation between the CLL cell count in peripheral blood and the clinical response to ibrutinib therapy is generally low[20], and there is an unmet need for early molecular markers of response to ibrutinib therapy.

Ibrutinib's molecular mechanism of action is rooted in the drug's inhibition of BTK, which results in downregulation of BCR signaling. Previous studies have investigated specific aspects of the molecular response to ibrutinib, for example investigating immunosuppressive mechanisms[23] and identifying decreased NF-κB signaling as a cause of reduced cellular proliferation[24–26]. However, a genome-scale, time-resolved analysis of the regulatory response to ibrutinib in primary patient samples has been lacking.

To dissect the precise cellular and molecular changes induced by ibrutinib therapy, and to identify candidate molecular markers of therapy response, here we follow individual patients with CLL ($n = 7$) at high temporal resolution (up to eight time points) over a standardized 240-day time course of ibrutinib treatment. Peripheral blood samples are analyzed for cell composition by flow cytometry, for epigenetic/regulatory cell state by ATAC-seq[27] on six different FACS-purified immune cell populations (158 ATAC-seq profiles in total), and for cell type specific transcriptional changes by single-cell RNA-seq[28] (scRNA-seq) applied to a subset of time points (>43,000 single-cell transcriptomes in total).

Integrative bioinformatic analysis of the resulting dataset identify a consistent regulatory program of ibrutinib-induced changes that is shared across all patients: Within the first days after the start of ibrutinib treatment, CLL cells display reduced NF-κB binding, followed by reduced activity of lineage-defining transcription factors, and erosion of CLL cell identity. Finally, after an extended period of ibrutinib treatment, CLL cells acquire a quiescence-like gene signature.

This drug-induced regulatory program is present in all patients, and we are able to validate it in an independent CLL cohort. We further observe substantial patient-to-patient variation in the speed with which these consecutive events unfold. Taking advantage of our time series data, we identify predictors of the time it takes for each patient to acquire an ibrutinib-induced molecular response, some of which were detectable already in pre-treatment samples.

In aggregate, our study provides a comprehensive, time-resolved analysis of the molecular and cellular dynamics upon ibrutinib treatment in CLL. It constitutes one of the first high-resolution, multi-omics time series of the molecular response to targeted therapy in cancer patients. The study also establishes a broadly applicable approach for analyzing drug-induced regulatory programs and identifying molecular response markers for targeted therapy. Importantly, the study's high temporal resolution with three complementary assays provides robust and informative results based on a small patient cohort. The presented approach may therefore be particularly relevant for obtaining maximum insight from early-stage clinical trials and cases of experimental off-label drug use that are intrinsically limited to few individuals.

## Results

**Ibrutinib therapy induces broad changes among immune cells.** To investigate the cellular dynamics and regulatory program induced by the inhibition of BCR signaling in CLL patients, we followed seven individuals from the start of ibrutinib therapy over a standardized time course of 240 days (Fig. 1a). All patients received the same treatment regimen with daily doses of ibrutinib and underwent extensive clinical monitoring. The patients covered a range of different demographic, clinical, and genetic parameters, representative of the spectrum of refractory CLL encountered in clinical practice (Supplementary Data 1).

For all patients and up to eight time points (0, 1, 2, 3, 8, 30, 120/150, 240 days after the start of ibrutinib therapy), we performed immunophenotyping by flow cytometric analysis of peripheral blood mononuclear cells (PBMCs), systematically quantifying changes in cell composition in response to ibrutinib therapy (Supplementary Fig. 1a and Supplementary Data 2). A gradual decrease in the percentage of CLL cells was observed over time (Fig. 1b), but with extensive temporal heterogeneity across patients (Supplementary Fig. 1b, c). The progressive reduction in CLL cells coincided with an increase in the percentage of non-malignant natural killer (NK) and T cell populations, consistent with a recent report[21]. This trend was most visible for CD8+ T cells (Fig. 1b, c and Supplementary Data 2), while CD4+ T cells remained largely unaffected. Although these differences were not statistically significant due to small cohort size, they were consistent with published data and provided both characterization and validation of our patient cohort.

Based on flow cytometry, we further observed a statistically significant loss of CLL-associated surface receptors (CD5, CD38) at the protein level specific to CLL cells, indicative of regulatory changes in CLL cells upon ibrutinib treatment (Fig. 1d, Supplementary Fig. 2, and Supplementary Data 3). For a systematic analysis of the ibrutinib-induced changes in gene expression, we performed single-cell RNA-seq[28] on the total PBMC population for a subset of patients and time points (Supplementary Data 4), capturing both the transcriptomes of CLL cells and of matched non-malignant immune cells. Overall, ~43,000 single-cell transcriptomes passed quality control (Supplementary Fig. 3a, b) and were integrated into a two-dimensional map using the UMAP method for unsupervised dimensionality reduction (Fig. 1e).

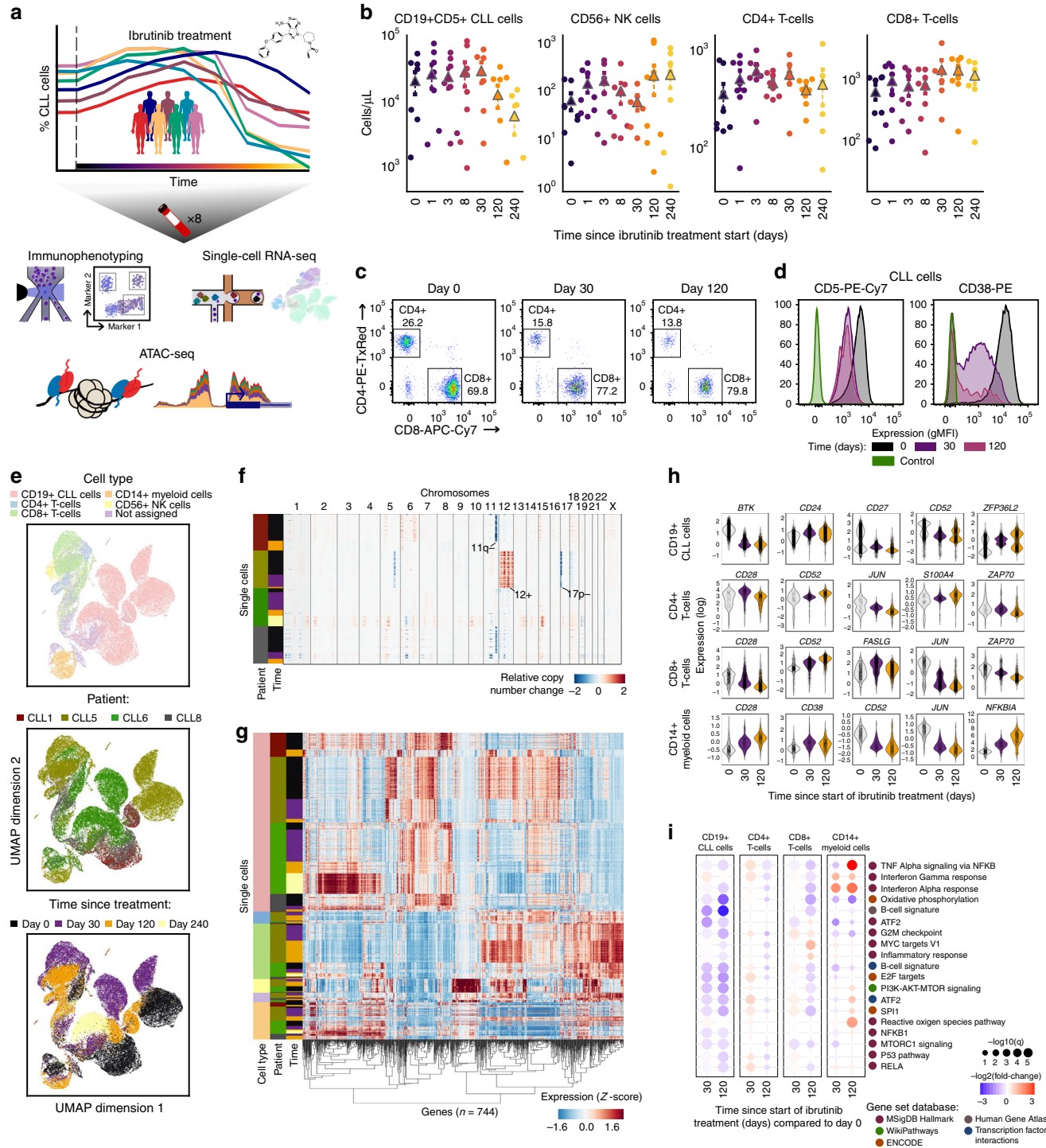

Cell type specific marker genes such as *CD79A*, *CD3D*, *CD14*, and *NKG7* were clearly detectable in the single-cell RNA-seq data and largely unaffected by ibrutinib treatment (Supplementary Fig. 3c), thus allowing for robust marker-based assignment of cell types. Cell counts inferred from scRNA-seq were almost perfectly correlated with those obtained by flow cytometry (Spearman's $\rho = 0.95$, Supplementary Fig. 3d), which provided independent validation of our dataset and of the analytical approach. Moreover, based on scRNA-seq data we were able to infer patient-specific copy number aberrations (Fig. 1f), which identified characteristic CLL-specific chromosomal aberrations

including deletion of chromosome 11q and 17p, and trisomy of chromosome 12.

Comparing the single-cell transcriptomes for each sample and cell type to the patient's corresponding pre-treatment (day 0) sample (Supplementary Fig. 3e–j and Supplementary Data 5), we found cell type specific trends in the molecular response to ibrutinib therapy (Fig. 1g, h and Supplementary Fig. 4). In CLL cells, we observed reduced expression of the ibrutinib target *BTK*, of *CD52* (a CLL disease activity marker[29]), and of *CD27* (a regulator of B-cell activation[30]). Among the non-malignant immune cell types, CD8$^+$ T cells were most strongly affected,

**Fig. 1 Multi-omics analysis of ibrutinib time courses reveals broad changes among immune cells. a** Schematic representation of the study design. Peripheral blood from patients with CLL undergoing single-agent ibrutinib therapy was collected at defined time points and assayed by flow cytometry (cell composition and immunophenotype), single-cell RNA-seq (gene expression), and ATAC-seq (chromatin accessibility). **b** Cell type abundance over the ibrutinib time course, as measured by flow cytometry. Triangles represent the mean for each time point and dashed lines indicate the 75% confidence interval around the mean, calculated across seven patients. **c** Flow cytometry scatterplots showing the abundance of T cell subsets for one representative patient at three time points (day 0: before the initiation of ibrutinib therapy, day 30 (120): 30 (120) days after the initiation of ibrutinib therapy). Cells positive for CD3 or CD8 were gated as indicated by the black rectangles and quantified as percentages of live PBMCs. **d** Flow cytometry histograms showing CD5 and CD38 expression on CLL cells (pre-gated for live, single CD19+CD5+ cells) for a representative patient and three time points. **e** Two-dimensional similarity map (UMAP projection) showing all 43,049 single-cell transcriptome profiles that passed quality control. Cells are color-coded according to their assigned cell types based on the expression of known marker genes. **f** DNA copy number profiles for CLL cells, as inferred from single-cell RNA-seq data. Three genetic aberrations common in CLL are indicated. For illustration, 2500 randomly selected CLL cells are shown for each patient. **g** Clustered single-cell transcriptome heatmap for the most differentially expressed genes between time points. For illustration, 20,000 randomly selected from a total of 43,049 cells are displayed. **h** Violin plots showing the distribution of gene expression levels for selected differentially expressed genes over the time course. **i** Differential gene expression signatures in four cell types, comparing each sample to the matched pre-treatment sample and averaging across patients. **e–g, i** Based on scRNA-seq data for 12 samples obtained from four patients.

which included downregulation of genes important for immune cell activation such as *CD28*, *JUN*, and *ZAP70*. This pattern was shared to a lesser extent by CD4+ T cells, while CD14+ cells showed strong upregulation of the NF-κB regulator *NFKBIA*.

Looking beyond individual genes, we characterized the response to ibrutinib by quantifying the transcriptome dynamics of predefined gene sets and transcriptional modules relevant to CLL and immunity (Fig. 1i and Supplementary Figs. 5, 6). We observed robust downregulation of B cell specific genes in CLL cells, including target gene sets of NF-κB subunits RELA and NF-κB1, and of the NF-κB associated transcription factors ATF2 and SPI1/PU.1. Genes involved in oxidative phosphorylation were also downregulated, consistent with widespread dampening of cellular activities in CLL cells under ibrutinib therapy. Among the non-malignant immune cell types, CD8+ T cells showed broad downregulation that was less pronounced but similar to the response observed in CLL cells, and CD14+ monocytes/macrophages showed specific upregulation of inflammatory response signatures including interferon gamma, TNF, and NF-κB signaling.

In summary, immunophenotyping and scRNA-seq over a dense time course of ibrutinib therapy uncovered widespread changes not only in CLL cells, but also in non-malignant immune cells. Most notably, we observed downregulation of NF-κB signaling and loss of B-cell surface markers in CLL, suggesting these are key contributors to the progressive reduction of the CLL cell fraction over time, and we observed a surprising degree of transcriptional change in non-CLL immune cells concomitant with an increase in the CD8+ T cell fraction.

**ATAC-seq uncovers an ibrutinib-induced regulatory program.** To dissect the regulatory basis of the ibrutinib-induced changes in the CLL cell transcriptomes and immunophenotypes, we performed ATAC-seq on FACS-purified CD19+CD5+ cells over the ibrutinib time course (Fig. 2a, Supplementary Fig. 7, and Supplementary Data 6). We modeled the temporal progression as Gaussian processes (a statistical method for handling time series data[31]) and identified 6797 genomic regions that underwent significant changes in chromatin accessibility in response to ibrutinib (Supplementary Data 7). Four major clusters were detected among these genomic regions (Fig. 2b): (i) regions that gradually lost chromatin accessibility ($n = 3412$); (ii) regions that gradually gained chromatin accessibility ($n = 2199$); (iii) regions that followed a bimodal, oscillating pattern ($n = 369$); and (iv) regions characterized by a peak in chromatin accessibility around 30 days after the start of ibrutinib treatment ($n = 354$).

We inferred the putative regulatory roles of these four clusters by region set enrichment analysis using the LOLA software[32]

(Fig. 2c). Cluster 1 (decrease in chromatin accessibility) was strongly enriched for binding sites of transcription factors with a role in lymphoid differentiation and gene regulation, and for enhancers specific to CLL cells and/or B cells. Cluster 2 (increase in chromatin accessibility) was enriched for B cell and T cell specific enhancers. Cluster 3 (bimodal, oscillating chromatin accessibility) was enriched for NF-κB binding sites. Lastly, cluster 4 (peak in chromatin accessibility around day 30) was enriched for transcribed regions marked by histone H3K36me3 in hematopoietic cells.

To identify potential regulators of the ibrutinib-induced modulation of CLL cell state, we focused on the enriched transcription factors (from Fig. 2c) and estimated their change in global binding activity over the ibrutinib time course, aggregating the ATAC-seq signal across each factor's putative binding sites (based on publicly available ChIP-seq data). As expected, several key transcription factors involved in B cell development (including NF-κB and PAX5) and B cell proliferation (including MEF2C and FOXM1) showed marked reduction of chromatin accessibility at their binding sites (Fig. 2d and Supplementary Fig. 8a). This effect was shared between CLL cells and non-malignant B cells, while it was not detected in other immune cell types.

Integrative analysis of chromatin accessibility and cell type specific transcription further refined this picture. When we performed parallel enrichment analysis for transcription factors and their putative binding sites (Fig. 2e), we observed concerted changes for key regulators of B cell development such as BCL11A, EBF1, IKZF1, IRF4, MEF2A, NFATC1, PAX5, and POU2F2, indicating that BTK inhibition may trigger loss of B cell identity in CLL cells. In support of this interpretation, we found global B cell specific gene expression signatures consistently downregulated upon ibrutinib treatment in CLL cells (Fig. 2f and Supplementary Fig. 8b).

Taken together, these results define a characteristic temporal order in which the ibrutinib-induced regulatory changes in CLL cells unfold. Already after one day of ibrutinib treatment, CLL cells showed reduced chromatin accessibility at NF-κB binding sites. This was followed by a gradual decrease in chromatin accessibility at binding sites of transcription factors that NF-κB regulates (PU.1[33], IRF4[34,35]) or that interact with NF-κB (ATF2[36]). Moreover, we observed reduced B cell specific regulatory activity, including decreased chromatin accessibility at regulatory regions specific to B cells and at the binding sites of B cell transcription factors such BCL11A, NFATC1, and RUNX3. These results highlight NF-κB mediated loss of B cell identity as the central regulatory change in CLL cells of patients undergoing ibrutinib therapy.

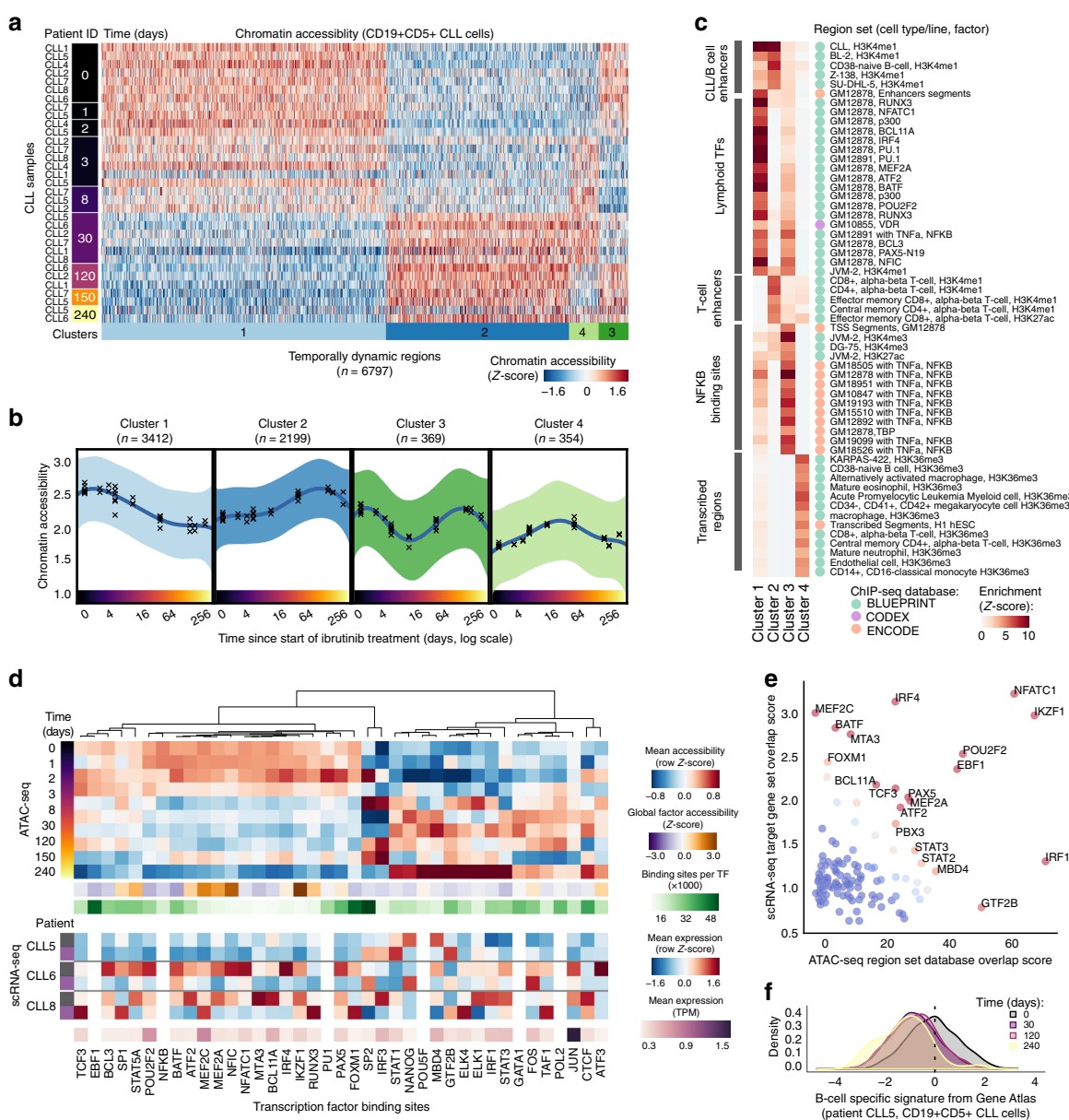

**Fig. 2 Changes in chromatin accessibility define an ibrutinib-induced regulatory program in CLL cells. a** Heatmap showing changes in chromatin accessibility for CLL cells over the time course of ibrutinib treatment, based on ATAC-seq data for 33 samples obtained from seven patients. **b** Mean chromatin accessibility across patients plotted over the ibrutinib time course in dynamically changing regulatory regions. Crosses represent samples from a single patient at a specific time point, and 95% confidence intervals are shown as colored shapes. **c** Region set enrichments for clusters of dynamic regions, calculated using the LOLA software. Enrichment p-values were Z-score transformed per column. **d** Heatmaps showing mean chromatin accessibility of regulatory regions overlapping with putative binding sites, expression of the corresponding transcription factor, and total number of its binding sites. Clustering was performed on the mean chromatin accessibility values. **e** Scatterplot showing differential regulation of transcription factors upon ibrutinib treatment. The x-axis displays the enrichment of transcription factors enriched in the LOLA analysis, and the y-axis displays the enrichment of their target genes among the differentially expressed genes. **f** Gene expression histogram across CLL cells in one patient, demonstrating the decline of a B cell-specific expression signature over the time course of ibrutinib therapy. For illustration, data are shown for the patient with most time points in the single-cell RNA-seq analysis (CLL5).

**Immune cell subsets acquire a quiescence-like gene signature**.
To characterize the effect of ibrutinib therapy on gene regulation in non-malignant immune cells, we performed ATAC-seq on FACS-purified CD19$^+$CD5$^-$ B cells, CD3$^+$CD4$^+$ T helper cells, CD3$^+$CD8$^+$ cytotoxic T cells, CD56$^+$ NK cells, and CD14$^+$ monocytes/macrophages from the same patients and time points (Supplementary Data 8). We identified a total of 12,574 temporally dynamic regulatory regions in these five cell types (Fig. 3a, b, Supplementary Fig. 9, and Supplementary Data 9).

Unsupervised clustering detected shared temporal dynamics across the five types of non-malignant immune cells, with sets of regions showing gradually decreasing or increasing chromatin accessibility over time, and a bimodal, wave-like cluster that was characterized by an initial decrease followed by a subsequent increase in chromatin accessibility (Fig. 3c and Supplementary Fig. 9a–c). Despite these shared temporal dynamics, the contributing genomic regions were highly cell type specific (Supplementary Fig. 9d), suggesting that the different immune cell types react in

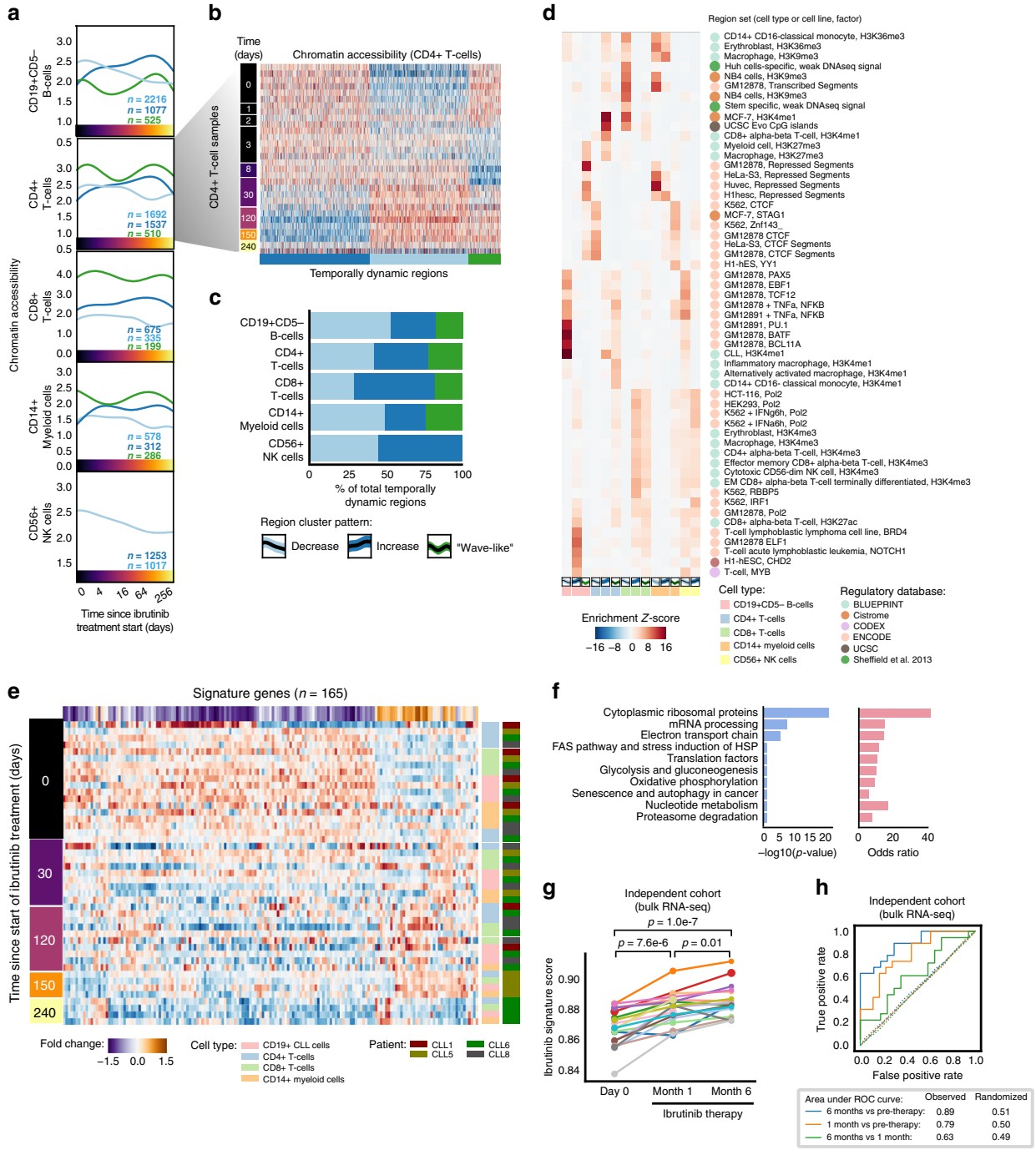

**Fig. 3 Non-malignant immune cells acquire a shared quiescence-like gene signature upon ibrutinib therapy. a** Mean chromatin accessibility across patients plotted over the ibrutinib time course for clusters of dynamically changing regulatory regions in five immune cell types, based on ATAC-seq data for 122 samples obtained from seven patients. **b** Heatmap of chromatin accessibility for CD4+ cells, illustrating dynamic regulation over the ibrutinib time course. **c** Stacked bar plots indicating the percentage of dynamically changing regions in each cluster. **d** Region set enrichments for the clusters of dynamically changing regions, calculated using the LOLA software and publicly available region sets as reference (mainly based on ChIP-seq data). Enrichment p-values were Z-score transformed per column. **e** Heatmap showing mean expression levels for genes that were differentially expressed over the ibrutinib time course when combining the data for CLL cells and for the five non-malignant immune cell types. Values represent column Z-scores of gene expression. **f** Gene set enrichments for genes downregulated across cell types, using WikiPathways as reference (Fisher's exact test, left: FDR-corrected p-value, right: odds ratio as a measure of effect size). **g** Expression score for the quiescence-like gene signature (as shown in **e**) in an independent cohort, calculated from bulk RNA-seq data for PBMCs collected before the start of ibrutinib therapy and at two subsequent time points. Significance was assessed using a paired t-test. **h** ROC curves illustrating the prediction performance of the gene signature (from **e**) for classifying samples in the independent validation cohort (solid lines). As negative controls, each prediction was repeated 100 times with permuted class labels, and the mean ROC curves across iterations are shown (dotted lines).

characteristic ways to the direct and indirect effects of ibrutinib treatment.

Among the non-malignant immune cell types, $CD19^+CD5^-$ B cells were most strongly affected by ibrutinib therapy, consistent with the central role of BCR signaling and of the ibrutinib target BTK for normal B cell function. Regions with decreasing chromatin accessibility in non-malignant B cells were enriched for similar transcription factor binding as CLL cells, and to a lesser extent also for NF-κB binding sites (Fig. 3d). We detected fewer regions with increasing chromatin accessibility upon ibrutinib treatment, and those regions lacked distinctive functional enrichment, suggesting that they are indirect effects downstream of the cells' direct response to ibrutinib treatment (Supplementary Data 9).

Ibrutinib-induced changes were not restricted to B cells. For example, regions with decreasing chromatin accessibility in $CD4^+$ T cells were enriched for binding sites of CTCF and RAD21, which are involved in three-dimensional chromatin organization; and regions with decreasing chromatin accessibility in $CD8^+$ T cells were enriched for histone marks associated with repressed chromatin in other cell types (Fig. 3d). Conversely, regions with increasing chromatin accessibility in $CD4^+$ T cells were enriched for interferon signaling and open, promoter-associated chromatin in T cells, while the enrichment observed for $CD8^+$ T cells included CpG islands and H3K4me1-marked regulatory regions (Fig. 3d).

Comparing the scRNA-seq data across cell types, we identified a characteristic set of genes that underwent similar transcriptional changes in CLL cells and in non-malignant immune cells (Fig. 3e and Supplementary Fig. 9e). This shared ibrutinib response signature was enriched for genes involved in ribosomal functions, mRNA processing, oxidative phosphorylation/metabolism, translation factors, senescence, and autophagy (Fig. 3f). For example, the shared ibrutinib response signature included *CD44*, a panlymphocyte cell adhesion molecule; *CD99*, a regulator of leukocyte migration, T cell adhesion, and cell death; *CD37*, which mediates the interaction of B and T cells; various surface proteins involved in cell adhesion (*CD52*, *CD164*, *ICAM3*, and *ITGB7*); the protein tyrosine kinase FGR, which is a negative regulator of cell migration; *TPT1*, a regulator of cellular growth and proliferation; and several factors involved in protein translation (*EEF2*, *EID1*, *EIF1*, and *EIF3E*) as well as ribosomal proteins (Supplementary Fig. 10a).

Interestingly, the shared ibrutinib response signature comprised genes that are involved in senescence and/or quiescence, including *CXCR4*, a chemokine receptor required for hematopoietic stem cell quiescence[37,38]; *ZFP36L2*, an RNA binding protein that promotes quiescence in developing B cells[39]; and *HMGB2*, a chromatin protein involved in the regulation of gene expression in senescent cells[40] (Supplementary Fig. 10). Our data thus suggest that CLL cells and non-malignant immune cells respond to ibrutinib therapy with shared transcriptional changes, including downregulation of genes involved in leukocyte function and cell–cell interactions, and upregulation of genes involved in quiescence and senescence.

To assess the reproducibility of this shared ibrutinib response signature in an independent validation cohort, we utilized recently published bulk RNA-seq data for PBMCs from patients with CLL ($n = 19$) that underwent single-agent ibrutinib treatment at a different medical center[26]. We indeed observed consistent changes in the expression of our gene signature for the vast majority of patients from the validation cohort (Fig. 3g). The difference was statistically significant at both time points compared to day 0 (month 1: $p = 7.6e-6$; month 6: $p = 1.0e-7$; paired *t*-test), and an accurate distinction was possible between patient samples collected before and during ibrutinib therapy

(receiver operating characteristic area under curve values of 0.89 and 0.79, respectively) (Fig. 3h).

In summary, our data show that ibrutinib therapy induces time-dependent regulatory changes not only in CLL cells but also in other immune cell types. Changes in non-malignant B cells mirrored those in CLL cells (albeit with a weaker NF-κB signature), while $CD4^+$ T cells, $CD8^+$ T cells, NK cells, and myeloid cells responded in cell type specific ways. Moreover, we identified and validated a gene expression signature that captures broad ibrutinib-induced downregulation of immune cell functions and acquisition of a quiescent-like state in response to ibrutinib therapy.

**Patient heterogeneity predicts response to ibrutinib therapy.** Our dataset and analyses clearly support the existence of an ibrutinib-induced regulatory program that is consistent across patients. Nevertheless, we also observed substantial patient-to-patient variability at the genetic (Fig. 1f), transcriptional (Fig. 3g), chromatin-regulatory (Fig. 2d), and cellular level (Supplementary Fig. 1b). Such heterogeneity in the presence of a shared regulatory program could be explained by patient-to-patient differences in the speed of progression through the program. If this is true, it could provide us with an opportunity to monitor or predict, based on molecular profiles, which patients pursue a faster or slower time course toward a sustained cellular response upon ibrutinib therapy.

We first investigated genetic heterogeneity over the ibrutinib time course, using copy number profiles inferred from the scRNA-seq data. This analysis identified changes in the subclonal composition of CLL cells over time within patients (Supplementary Fig. 10a–d). We did not observe a strong correlation between individual copy number aberrations and our single-cell "ibrutinib molecular response score", calculated as the expression intensity of our validated ibrutinib response signature (Fig. 3e) in individual CLL cells based on their scRNA-seq profiles (Supplementary Fig. 10e–h). However, we did observe an association between the subclonal genetic heterogeneity over the time course of ibrutinib treatment, quantified by a measure that we validated on simulated data and on the changing ratio of CLL cells versus non-malignant cells upon ibrutinib treatment (Fig. 4a and Supplementary Fig. 10i–l), and the speed of the cellular response to ibrutinib treatment as measured by flow cytometry (Fig. 4b). This finding suggests that, over the time course of 120 days covered by our study, ibrutinib effectively depletes the CLL majority clone in the most rapidly responding patients, thereby unmasking subclonal genetic heterogeneity present in these patients.

Second, we investigated the association of chromatin accessibility in CLL cells at day 0 with a range of patient-specific characteristics. To that end, we performed principal component analysis on the chromatin profiles for all patients and cell types, and we tested for statistical associations with clinical annotations (Supplementary Fig. 11a). We observed a strong association between the second principal component of the chromatin profiles in CLL cells at day 0 and the cellular response to ibrutinib treatment at day 120, suggesting that this chromatin signature provides an epigenomic marker for the subsequent cellular response to ibrutinib therapy (Fig. 4c, d). This chromatin signature separated patients into fast versus slow responders to ibrutinib therapy independently of other clinical annotations (Supplementary Fig. 11b). Genomic regions associated with a slow response to ibrutinib therapy showed similar enrichments as regions with reduced chromatin accessibility in CLL cells (Fig. 2c), including preferential overlap with broadly active enhancer regions and transcription factor binding sites (Supplementary Fig. 11c). Moreover, we observed specific enrichment

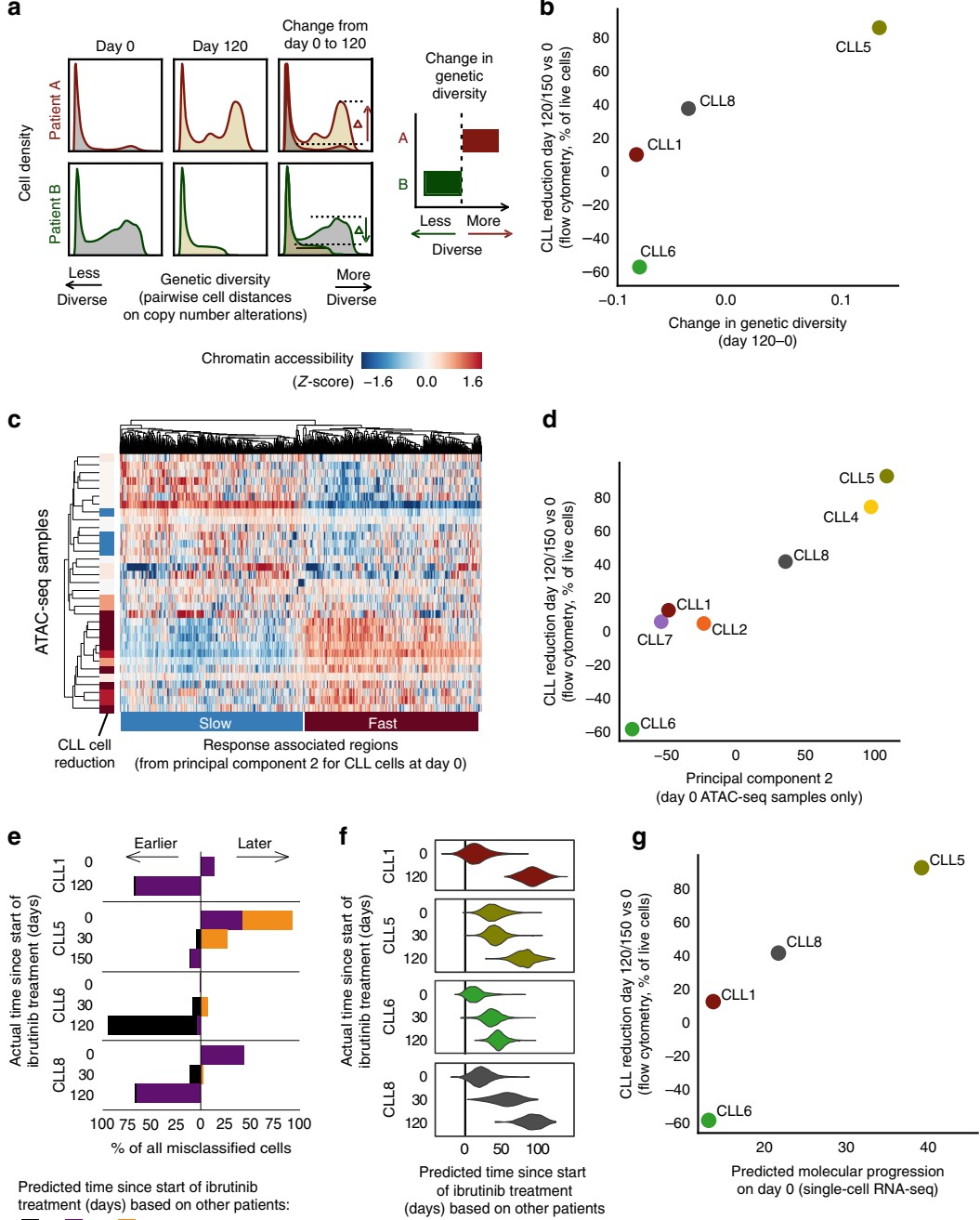

**Fig. 4 Heterogeneity across patients reflects and predicts the patient-specific temporal response to ibrutinib. a** Computational approach to quantify changes in genetic diversity based on copy number profiles inferred from the single-cell RNA-seq data. Shifts in the distribution of pairwise distance similarities between time points indicate changes in the genetic diversity of the cell population. **b** Scatterplot comparing across patients the change in genetic diversity between day 0 and day 120/150 of ibrutinib treatment (x-axis) with the change in the CLL cell percentage on day 120/150 of ibrutinib treatment compared to day 0 as measured by flow cytometry (y-axis). **c** Clustered heatmap for chromatin accessibility profiles of CLL cells, based on ATAC-seq data for 33 samples obtained from seven patients. The heatmap shows the top 1000 genomic regions that at day 0 associate with the second principal component (Supplementary Fig. 11a), annotated on the left with the change in CLL cell fraction (as in **b**). **d** Scatterplot comparing across patients the average chromatin accessibility for regions linked to the second principal component (as in **c**, x-axis) with the change in CLL cell fraction (as in **b**, y-axis). **e** Stacked bar charts showing the number and direction of deviations from the actual collection time point when predicting time points in each patient after training the classifier in all other patients. **f** Violin plots showing the predicted (x-axis) and actual (y-axis) number of days under ibrutinib therapy in each patient. Predictions are derived from regression models trained on all other patients. **g** Scatterplot comparing the predicted time under ibrutinib therapy (from **f**, x-axis) with the change in CLL cell fraction (as in **b**, y-axis).

for gene sets associated with metabolic activity, consistent with reports that ibrutinib-resistant cells exhibit a distinct metabolic state compared to sensitive cells[41,42].

Third, based on the scRNA-seq data we derived and evaluated gene expression signatures that capture the molecular response to ibrutinib treatment in individual cells. Using machine learning, we predicted the time of sample collection (day 0, 30, or 120/150) for each of the ~19,000 single-cell transcriptomes for CLL cells from four donors. Both support vector machines and elastic net classifiers achieved excellent prediction performance with cross-validated test set ROC area under curve (AUC) values in the range of 0.975 to 0.999, and these results were robust to differences in the number of detected genes across cells (Supplementary Fig. 12a). Our results indicate that the transcriptome profiles of single CLL cells undergo changes that reflect the duration of ibrutinib therapy—a finding that may be exploited for molecular staging of patient-specific ibrutinib responses. Using a classifier that was trained and evaluated by patient-stratified cross-validation, we observed that cells from specific patients were consistently predicted to have progressed faster (CLL5) or slower (CLL6) along the trajectory of the transcriptional response to ibrutinib treatment (Fig. 4e), indicating that individual patients indeed follow their own timelines in the molecular response to ibrutinib therapy.

Finally, for a more quantitative assessment of these temporal dynamics, we trained and evaluated regression models that predict the precise time (i.e., number of days) after the start of ibrutinib therapy for each individual CLL cell transcriptome. We observed excellent test set prediction performance for three patients (CLL1, CLL6, and CLL8), with $r^2$ values (i.e., percent variance explained) of 92.3, 84.2, and 78.1%, respectively (Supplementary Fig. 12b, c). Lower performance was observed for CLL5 ($r^2 = 36.6\%$), where the day-0 time point already showed a signature reminiscent of ibrutinib treatment (Fig. 4f). To interpret the fitted models, we compared the regression coefficients across patients (Supplementary Fig. 13a, b). This analysis identified gene sets that were consistently associated with faster or slower reduction of CLL cells at day 120 (Supplementary Fig. 13c–g). For example, expression levels of the transcription factor JUN, the translation initiation factor EIF1, the ubiquitin protein UBC, the interleukin 4 receptor IL4R, and the MHC class II co-receptor CD74 were associated with slow reduction of CLL cell numbers, while the p53 cofactor DDX5, the TNF receptor protein CD27, and the signaling protein CD24 were associated with fast reduction. Consistent with the classification analysis (Fig. 4e), our regression models predicted individual patients progressing faster (CLL5) or slower (CLL6) along the trajectory of the transcriptional response to ibrutinib, while the two remaining patients (CLL1, CLL8) followed similar timelines (Fig. 4f). When we compared predictions based on CLL single-cell transcriptomes at day 0 across patients, we found that the observed molecular signature prior to the start of ibrutinib treatment indeed anticipated the subsequent cellular response (i.e., reduction of CLL cells on day 120/150 compared to day 0) (Fig. 4g).

These results indicate that genetic, epigenetic, and transcriptional heterogeneity across patients captures inter-individual differences in the response to ibrutinib treatment. The different molecular profiling technologies applied in this study may provide complementary sources of candidate markers that can predict the patient-specific time until a strong cellular response to ibrutinib treatment is achieved for individual patients with CLL.

## Discussion

Multi-omics analysis of clinical time courses has emerged as a powerful method for dissecting the molecular response to targeted therapy, allowing us to define the precise temporal order of induced molecular changes and to unravel underlying regulatory programs. Here, we applied flow cytometry, scRNA-seq, and chromatin mapping in six FACS-purified cell types to a dense time course of patients with CLL who are starting ibrutinib therapy. These three assays provide comprehensive and complementary information comprising cell composition and immunophenotypes (flow cytometry), transcriptional changes across all major immune cell populations (scRNA-seq), and chromatin dynamics that may explain and predict the observed changes in transcription regulation and epigenetic cell state (ATAC-seq).

Integrative bioinformatic analysis identified a characteristic regulatory program that was shared across all patients. Among the earliest changes following the start of ibrutinib therapy, we observed a decrease of NF-κB binding signatures in CLL cells, which was followed by a rapid reduction in the regulatory activity of transcription factors involved in B cell development and function (such as EBF1, FOXM1, IRF4, PAX5, and PU.1). This decrease was accompanied by (and may indeed cause) the downregulation of CLL-specific gene signatures and a decrease in surface marker levels such as CD5 and CD19, together indicating a broad erosion of CLL cell identity.

Ibrutinib-induced changes were not exclusive to CLL cells but shared with other immune cell types. Non-malignant B cells largely mirrored the changes observed in CLL cells—which was expected given the important role of the ibrutinib target BTK in BCR signaling. We further observed a dampening effect of ibrutinib on immune pathway regulation in CD8+ T cells, while there was an increase of inflammatory gene signatures in monocytes/macrophages. These changes in immune cell types that do not express BTK could be due to a combination of direct effects via ibrutinib's promiscuous inhibition of kinases other than BTK (including BLK, BMX, ITK, TEC, TXK, and EGFR[43]) and indirect effects arising from the ibrutinib-induced relocation of CLL cells from the protective microenvironment into the peripheral blood.

Interestingly, both for CLL cells and for non-malignant immune cells, sustained ibrutinib therapy eventually resulted in the acquisition of a shared, quiescence-like gene signature. We successfully validated our gene signature in an independent clinical cohort of patients with CLL who start ibrutinib treatment. This gene signature may help explain certain cellular and clinical phenotypes observed in patients under ibrutinib therapy, including changes in the immune microenvironment[44] and increased susceptibility to infections[45–47]. For example, downregulation of CD99 indicates that Fas-mediated T-cell death may be impaired[48,49], which has been proposed as a cause of CD8+ T cell accumulation in peripheral blood[21]. Moreover, two genes in the signature (CXCR4 and ZFP36L2) have established biological functions in senescence and quiescence of hematopoietic cells[37,38]. Moreover, ibrutinib is known to inhibit CXCR4-mediated expression of CD20 in CLL cells[50], which could have implications for clinical trials combining ibrutinib and anti-CD20 antibodies (e.g., NCT02007044).

Our comprehensive, time-resolved, multi-omics analysis of ibrutinib therapy provides integration and context for previous studies that have focused on specific aspects of the response to ibrutinib, including reduced proliferation[25], decreased cell-cell contacts[21,22], and downregulation of NF-κB[24–26]. Moreover, the identification of a regulatory program that was shared across all patients allowed us to explore patient-to-patient heterogeneity in the speed with which this program is executed, suggesting that it may be possible to define predictive molecular markers for the cellular response to ibrutinib therapy. Most notably, our chromatin analysis identified a patient-specific signature present prior

to treatment that correlated with the speed of CLL cell clearance, and our scRNA-seq data predicted the cellular response measured 120/150 days after the start of ibrutinib therapy. Our results capture complementary aspects of CLL biology and demonstrate the power of combined multi-omics profiling for patient-specific treatment monitoring. While these findings remain exploratory due to the small number of patients in our study, they raise the future perspective of quantifying and predicting the molecular response for a growing class of targeted cancer therapies that are not primarily cytotoxic and for which simple cell-based biomarkers (e.g., leukemic cell count or minimal residual disease) are poor predictors of clinical response. Moreover, when combined with our recent analysis of chromatin profiles and single-cell drug responses in CLL[51], the opportunity arises for deriving patient-specific, mechanistically justified, combination therapies for individual patients.

While we consider our approach broadly applicable in the context of precision oncology and targeted therapy, the following limitations apply: First, such comprehensive profiling (up to 8 time points, dozens of genome-wide ATAC-seq profiles, and thousands of single-cell transcriptomes per patient) is challenging to implement for large patient cohorts. Second, given the small size of our cohort, we could not systematically account for known genetic risk markers in CLL. However, recent studies have shown that established prognostic markers have lost much of their predictive power with ibrutinib therapy[16,52], and the same may apply to other emerging treatments such as CAR T cell therapy[53]. Third, while we found evidence of subclonal heterogeneity in our single-cell transcriptome data, the current throughput of scRNA-seq does not (yet) enable deep characterization of the subclonal architecture. Fourth, most patients that start ibrutinib therapy have previously been treated with other drugs (1–5 prior treatments in our cohort), which may explain some of the differences in the speed of the molecular response to ibrutinib. Fifth, time series data support only a weak form of causal inference (Granger causality[54,55]), where earlier events may cause later events but not vice versa (e.g., the observed decrease in NF-κB binding was followed by a downregulation of B cell transcription factors and an erosion of B cell identity among the CLL cells). Such results should therefore be considered causal in a strict biological sense only after mechanistic experimental validation in suitable disease models. When these limitations are taken into account, we expect that the presented approach will readily generalize to other targeted therapies, supporting the precise and robust definition of regulatory programs, molecular response monitoring, and identification of markers for the clinical response to therapy.

In conclusion, our study demonstrates the power of multi-omics and single-cell profiling, combined with integrative bioinformatic analysis, for dissecting the cellular, transcriptional, transcription-regulatory, and epigenetic impact of targeted therapies. A key strength of this approach is the high level of detail and biological insight that can be obtained from a small number of patients, which makes it particularly well-suited for applications in personalized medicine where each patient may follow a different disease trajectory. Moreover, the approach appears promising for early-stage clinical trials of new targeted therapies, where it is critical to obtain a robust assessment of the induced molecular and cellular dynamics, in order to inform dose finding and to provide biomarker candidates for molecular response monitoring.

## Methods

**Sample acquisition and clinical data**. All patients were treated at the Department of Hematology and Stem Cell Transplantation, Central Hospital of Southern Pest, Budapest, Hungary, according to the revised guidelines of the International Workshop Chronic Lymphocytic Leukemia/National Cancer Institute[56]. The study complied with all relevant ethical regulations for working with patients and patient samples. Informed consent was obtained from all participants. The study was approved by the ethical committees of the contributing institutions (Dél-Pesti Centrumkórház, Semmelweis University, and Medical University of Vienna).

**Flow cytometry and FACS**. Patient PBMCs were thawed and washed twice with PBS containing 0.1% BSA and 5 mM EDTA (PBS+BSA+EDTA). Cells were then incubated with anti-CD16/CD32 (clone 93, 1:200, Cat# 101301) to prevent non-specific binding. Single-cell suspensions were stained with combinations of antibodies against CD3 (FITC, clone UCHT1, 1:200, Cat# 300452), CD4 (PE-TxRed, clone OKT4, 1:200, Cat# 317448), CD5 (PE-Cy7, clone UCHT2, 1:100, Cat# 300622), CD8 (APC-Cy7, clone SK1, 1:150, Cat# 344746), CD14 (PerCp-Cy5.5, clone M5E2, 1:100, Cat# 301824), CD19 (APC, clone HIB19, 1:100, Cat# 302212), CD25 (PE-Cy7, clone BC96, 1:100, Cat# 302612), CD38 (PE, clone HB-7, 1:100, Cat# 356604), CD45RA (PerCp-Cy5.5, clone HI100, 1:100, Cat# 304122), CD45RO (AF700, clone 304218, 1:100, Cat# 304218), CD56 (AF700, clone NCAM16.2, 1:100, Cat# 340363, BD Bioscience), CD127 (APC, clone A019D5, 1:100, Cat# 351342), CD197 (CCR7, PE, clone G043H7, 1:100, Cat# 353204), and DAPI viability dye (all obtained from Biolegend unless stated otherwise) for 30 min at 4 °C followed by two washes with PBS+BSA+EDTA. For flow cytometry, cells were acquired with an LSRFortessa Cell Analyzer (BD). For FACS, cells were sort-purified with a MoFlo Astrois (Beckman Coulter) using the gating strategy depicted in Supplementary Fig. 1a. Data analysis was performed with the FlowJo (Tree Star) software. In Fig. 1d, control cells in CD5-PE-Cy7 channel are CD14+ myeloid cells, and control cells in CD38-PE channel are CD3+ CD4-CD8- cells (these cell populations do not express the respective markers and are therefore used to estimate background levels).

**Droplet-based single-cell RNA-seq**. Single-cell libraries were generated using the Chromium Controller and Single Cell 3′ Library & Gel Bead Kit v2 (10× Genomics) according to the manufacturer's protocol. Briefly, an aliquot of patient PBMCs was stained with DAPI for discrimination between live and dead cells, and a maximum of 100,000 live, doublet-excluded cells were sorted into 1.5 ml tubes. Cells were pelleted by centrifuging for 5 min at 4 °C at 300 × g and resuspended in PBS with 0.04% BSA. Up to 17,000 cells suspended in reverse transcription reagents, along with gel beads, were segregated into aqueous nanoliter-scale Gel Beads in Emulsion (GEMs). The GEMs were then reverse-transcribed in a C1000 Thermal Cycler (Bio-Rad) programmed at 53 °C for 45 min, 85 °C for 5 min, and hold at 4 °C. After reverse transcription, single-cell droplets were broken, and the single-strand cDNA was isolated and cleaned with Cleanup Mix containing Dynabeads MyOne SILANE (Thermo Fisher Scientific). cDNA was then amplified with a C1000 Thermal Cycler programmed at 98 °C for 3 min, 10 cycles of (98 °C for 15 s, 67 °C for 20 s, 72 °C for 1 min), 72 °C for 1 min, and hold at 4 °C. Subsequently, the amplified cDNA was fragmented, end-repaired, A-tailed, and index adapter ligated, with cleanup in-between steps using SPRIselect Reagent Kit (Beckman Coulter). Post-ligation product was amplified with a T1000 Thermal Cycler programmed at 98 °C for 45 s, 10 cycles of (98 °C for 20 s, 54 °C for 30 s, 72 °C for 20 s), 72 °C for 1 min, and hold at 4 °C. The sequencing-ready library was cleaned up with SPRIselect beads and sequenced by the Biomedical Sequencing Facility at CeMM using the Illumina HiSeq 3000/4000 platform and the 75 bp paired-end configuration.

**Assay for transposase-accessible chromatin (ATAC-seq)**. For chromatin accessibility mapping, a maximum of 50,000 sorted cells were pelleted by centrifuging for 5 min at 4 °C at 300 × g. After centrifugation, the pellet was carefully resuspended in the transposase reaction mix (12.5 μl 2 × TD buffer, 2 μl TDE1 (Illumina), and 10.25 μl nuclease-free water, 0.25 μl 5% Digitonin (Sigma)) for 30 min at 37 °C. Following DNA purification with the MinElute kit eluting in 11 μl, 1 μl of the eluted DNA was used in a quantitative PCR reaction to estimate the optimum number of amplification cycles. Library amplification was followed by SPRI size selection to exclude fragments larger than 1200 bp. DNA concentration was measured with a Qubit fluorometer (Life Technologies). Library amplification was performed using custom Nextera primers[27]. The libraries were sequenced by the Biomedical Sequencing Facility at CeMM using the Illumina HiSeq 3000/4000 platform and the 50 bp single-read configuration.

**Preprocessing and analysis of single-cell RNA-seq data**. Preprocessing of the single-cell RNA-seq data was performed using Cell Ranger version 2.0.0 (10× Genomics). Raw sequencing files were demultiplexed using the Cell Ranger command "mkfastq". Each sample was aligned to the human reference genome assembly "refdata-cellranger-GRCh38-1.2.0" using the Cell Ranger command "count", and all samples were aggregated using the Cell Ranger command "aggr" without depth normalization. Raw expression data were then loaded into R version 3.4.0 and analyzed using the Seurat package version 2.0.1 with the parameters suggested by the developers[57]. Specifically, single-cell profiles with less than 200 detected genes (indicative of no cell in the droplet), more than 3000 detected genes (indicative of cell duplicates), or more than 15% of UMIs stemming from mitochondrial genes were discarded. Read counts were normalized dividing by the total UMI count in each cell, multiplied by a factor of 10,000, and log transformed. The

number of UMIs per cell and the percent of mitochondrial reads per cell were then regressed out using Seurat's standard analysis pipeline.

**Dimensionality reduction and analysis of gene expression**. Principal component analysis, t-SNE analysis, hierarchical clustering, and differential expression analyses were carried out in R, using the respective functions of the Seurat package. t-SNE and cluster analyses were based on the first ten principal components. A negative binomial distribution test was used for differential analysis on genes expressed in at least 10% of cells in one group. Results were aggregated across patients by taking the mean for log fold changes and by Fisher's method for $p$-values. Enrichment analyses were done using Enrichr API[58] against the following databases: Transcription Factor PPIs, ENCODE, ChEA Consensus TFs from ChIP-X, NCI-Nature 2016, WikiPathways 2016, Human Gene Atlas, and Chromosome Location. Aggregate gene expression values for gene sets (signatures) were quantified as follows: Log-normalized transcript per $10^4$ UMI counts were scaled between 0 and 1. The values for all genes of a given set were then summed to obtain a raw value for each gene set and cell. To remove cell specific effects such as differences in UMI distributions due to sequencing depth, raw values were transformed to Z-scores using a distribution of raw values of 500 randomly picked gene sets of the same size. Differences in signatures between time points were assessed using "$t$.test" in R. Results were aggregated across patients by taking the mean for log fold changes and by Fisher's method for $p$-values. Multiple testing correction of differentially expressed genes, enriched terms, and differences in signatures was carried out using the Benjamini-Hochberg procedure as implemented by the "p.adjust" function in R. The selected gene sets included 50 'hallmark signatures' from MSigDB[59], as well as ATF2, BATF, NFIC, NFKB1, RELA, RUNX3, and SPI target genes, and B cell signatures from Human Gene Atlas, NCI Nature 2016, and WikiPathways 2016, all obtained from Enrichr[58]. For data representation, we denoised the dataset with the Deep Count Autoencoder (DCA) in Python[60], using raw UMI counts as input and the "Zero-Inflated Negative Binomial" model (which explained the relationship between mean expression and observed dropout rates significantly better than the "Negative Binomial" model). The DCA-denoised data were then normalized per cell, log-transformed, and scaled. Dimensional reduction was performed by principal component analysis, and the resulting dimensions were used for neighbor graph construction followed by Uniform Manifold Approximation and Projection (UMAP) with Scanpy's default parameters[61].

**Preprocessing and analysis of ATAC-seq data**. ATAC-seq reads were trimmed using Skewer[62] and aligned to the GRCh37/hg19 assembly of the human genome using Bowtie2[63] with the "-very-sensitive" parameter. Duplicate reads were removed using the sambamba[64] "markdup" command, and reads with mapping quality >30 and alignment to the nuclear genome were kept. All downstream analyses were performed on these filtered reads. Peak calling was performed with MACS2[65] using the "-nomodel" and "-extsize 147" parameters, and peaks overlapping blacklisted features as defined by the ENCODE project[66] were discarded. We created a consensus region set by merging the called peaks from all samples across patients and cell types, and we quantified the accessibility of each region in each sample by counting the number of reads from the filtered BAM file that overlapped each region. To normalize the chromatin accessibility signal across samples, we first performed quantile normalization using the R implementation in the preprocessCore package ("normalize.quantiles" function). We then performed principal component analysis (scikit-learn, sklearn.decomposition.PCA implementation) on the normalized chromatin accessibility values of all chromatin-accessible regions across all samples. Upon inspection of the sample distribution along principal components, we noticed an association of several (but not all) samples from one processing batch with a specific principal component, while we did not observe any association of these samples with any known biological factor. To remove the effect of this latent variable while retaining variation from other (biological) sources, we performed principal component analysis on the matrix of raw counts on a per cell type basis (except myeloid cells, which contained no such samples) and removed the latent variable (first principal component) by subtracting the outer product of the transformed values of each sample in this component and the loadings of each regulatory element in the same component from the original matrix. We re-normalized the corrected count matrix and component analysis jointly for all cell types.

**Time series modeling of chromatin accessibility dynamics**. We modeled the temporal effect of ibrutinib in each cell type as a function of time by a latent process, which has emerged as a powerful approach for time series analysis[67–69]. To that end, we used the Python library GPy to fit Gaussian process regression models (GPy.models.GPRegression) on the log2 transformed sampling time on ibrutinib therapy (independent variable) and the normalized chromatin accessibility values for each regulatory element (dependent variable) for each cell type separately. We then fitted a variable radial basis function (RBF) kernel as well as a constant kernel (both with an added noise kernel), and we compared the log-likelihood and standard deviation of the posterior probability of the two as previously described[67–69]. Dynamic regulatory elements were defined as those for which the survival function of the chi-square of the D statistic (twice the difference between the log-likelihood of the variable fit minus the log-likelihood of the constant fit) was lower than 0.05 and the

standard deviation of the posterior was higher than 0.05. We then used the "mixture of hierarchical Gaussian process" (MOHGP) method to cluster regulatory elements according to their temporal pattern. The MOHGP class from the GPclust library (GPclust.MOHGP) was fitted with the same data as before, this time with a Matern52 kernel (GPy.kern.Matern52) and an initial guess of four region clusters. Regions with posterior probability higher than 0.8 were selected as dynamic and included in the downstream analysis.

**Region set enrichment analysis**. We performed region set enrichment analysis on the clusters of dynamic genomic regions using LOLA[32] and its core database, which comprises transcription factor binding sites from ENCODE[66], tissue-specific DNase hypersensitive sites[70], the CODEX database[71], UCSC Genome Browser annotation tracks[72], the Cistrome database[73], and data from the BLUEPRINT project[74]. Enrichment of genes associated with regulatory elements (annotated with the nearest transcription start site from Ensembl) was performed through the Enrichr API[58] for the following databases of gene sets: BioCarta 2016, ChEA 2016, Drug Perturbations from GEO down, Drug Perturbations from GEO up, ENCODE and ChEA Consensus TFs from ChIP-X, ENCODE TF ChIP-seq 2015, ESCAPE, GO Biological Process 2017b, GO Molecular Function 2017b, KEGG 2016, NCI-Nature 2016, Reactome 2016, Single Gene Perturbations from GEO down, Single Gene Perturbations from GEO up, and WikiPathways 2016.

**Inference of global transcription factor activity**. Global transcription factor accessibility was assessed by aggregating the normalized chromatin accessibility values of regulatory elements that overlap a consensus of regions (union of all sites) from ENCODE ChIP-seq peaks of the same factor across all cell types profiled. The mean accessibility of each sample in the sites overlapping binding sites of each factor was computed and subtracted by the mean accessibility of each sample across all measured regulatory elements. For visualization, we aggregated samples by cell type and sampling time point, displaying either the mean or a Z-score of chromatin accessibility. For all gene-level measures of chromatin accessibility, we used the mean of all regulatory elements associated with a gene, defined as the gene with the closest transcription start site as annotated by the RefSeq gene models for the hg19 genome assembly.

**Integrative analysis of ATAC-seq and scRNA-seq data**. To assess the agreement between the two analyses at the enrichment level, we performed enrichment analysis with Enrichr for genes differentially expressed across patients in the same cell type, and we compared the significance of terms for transcription factors in the "ENCODE TF ChIP-seq 2015" gene set library with the significance of transcription factors enriched in the LOLA analysis for each ATAC-seq cluster. To identify a common transcriptional signature associated with ibrutinib treatment across cell types, we selected all genes that were differentially expressed with the same direction in at least ten combinations of cell type and time point. These genes were split according to the direction of change with time and used for enrichment analysis with Enrichr as described above. The same genes were used to derive a score calculated as the mean expression of the upregulated genes over the mean expression of downregulated genes. An independent cohort of RNA-seq on bulk PBMCs from CLL patients[26] was used to assess the reproducibility of the signature by observing the significance of the difference between scores upon ibrutinib treatment with a paired-samples $t$-test. To assess the performance of the score as a classifier, we generated a ROC curve by counting true positive and negative rates with a sliding score threshold and calculated the area under the curve with scikit-learn's function "sklearn.metrics.auc".

**Inference of DNA copy number variation from scRNA-seq data**. To infer DNA copy number profiles at the single-cell level, we started with DCA-denoised, normalized, and scaled single RNA-seq data of all cells. We removed per-cell differences by subtracting the median expression of each cell from all genes and per-gene differences by subtracting the median and dividing by the standard deviation. We then calculated a rolling mean of expression across genes ordered by their chromosomal position for each chromosome individually. To improve the representation of DNA copy number profiles, we centered the resulting matrix by subtracting the mean of all values in the matrix and applied smoothing by cubing the matrix values (which shrinks small changes relative to all cells) and multiplying them by 3 (which scales values back to usual copy number variation bounds). Visual inspection of DNA copy number profiles identified copy number aberrations that are commonly observed in CLL. Moreover, for a subset of time points and patients, the results were validated by panel sequencing data for the same samples[75]. To discover clusters of genetically distinct cells within patients, we performed dimension reduction using principal component analysis on the smoothed matrix, computed a neighbor graph between cells, and fitted a UMAP manifold for the CLL cells of each sample (i.e., per patient and time point). This was overlaid with the response to ibrutinib of each single cell based on ibrutinib response signature described above. To assess global changes in genetic diversity within cells of a patient over time, we developed a global metric of genetic diversity based on inferred copy number profiles from single-cell RNA-seq data. We calculated pairwise Pearson correlation coefficients between all cells and used the square of the mean of this distribution as a measure of genetic diversity. To

benchmark this approach, we first established simulated copy number profiles with the same dimensions are the inferred one but for varying total numbers of cells. We created two populations where we simulated gain or loss of chromosome 12 (log change: −1 or 1) whereas the remainder of the genome was Gaussian noise of mean zero and standard deviation 0.1. We assessed performance by mixing the two populations together in different ratios and computing Pearson correlation between the population fraction (ground truth) and the predicted global diversity. An additional benchmark was performed by taking advantage of natural, known mixtures of cell types in the data. For these data, the inferred change in genetic diversity is simply the difference between global diversity measures between time points of ibrutinib treatment for each patient.

**Prediction of patient-specific response time from scRNA-seq**. The time point of sample collection (day 0, 30, or 120/150) for each CLL single-cell transcriptome was predicted using the glmnet package in R with a multinomial response variable (for classification) and the "alpha" parameter (lasso penalty) set to 1. Prediction performance was assessed by 3-fold cross-validation for each patient, where optimal "lambda" parameters were obtained separately for each (outer) fold in a 5-fold inner cross-validation using the function cv.glmnet. Parameter "lambda" for the final prediction across patients were obtained by 5-fold cross-validation on all data for each patient using cv.glmnet. Predictions were aggregated for each patient by taking the mean of dummy variables (1: early, 2: mid, 3: late) across the three other patients. Classification performance for support vector machines was assessed using the LiblineaR package in R. Classifiers were trained with "type" parameter 0 and "cost" parameters estimated by the heuristicC method on the training data, where cells were split ten times into 70% for training and 30% for testing. Quantitative prediction of the precise time (number of days) after the start of ibrutinib therapy was performed using the glmnet package in R with a Gaussian response variable (for regression) and the "alpha" parameter (lasso penalty) set to 1. Prediction performance was assessed using the "lambda" parameter that provided the highest $R^2$ in the training data of each fold. The regularization parameter "lambda" for the final prediction was obtained based on the mean squared error in a 3-fold cross-validation repeated five times on all data from each patient. Predictions were aggregated by taking the mean across three patients. For the analysis of regression coefficients, we retrieved these coefficients for each patient under the chosen "lambda" parameter, and we calculated summary statistics for each gene across patients. Based on these results, we identified the genes with high correlation to CLL cell reduction on day 120 (absolute Pearson correlation coefficient above 0.9), for which we illustrated the relationship between genes by calculating pairwise correlation matrices of coefficients across patients. In all Python analysis, we set the pseudo-random number generation seed state to 1142101101 in both the standard library "random" and in "numpy".

**Reporting summary**. Further information on research design is available in the Nature Research Reporting Summary linked to this article.

## Data availability
All data are available through the Supplementary Website (http://cll-timecourse. computational-epigenetics.org/). Single-cell RNA-seq and ATAC-seq data have been deposited in the NCBI GEO database and are publicly available under accession number GSE111015.

## Code availability
The analysis source code underlying the final version of the paper is openly accessible as a Git repository on Github (https://github.com/epigen/cll-ibrutinib_time).

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

## Acknowledgements

We thank all patients who have donated their samples for this study. We also thank the Biomedical Sequencing Facility at CeMM for assistance with next generation sequencing and all members of the Bock lab for their help and advice. N.F. is supported by a fellowship from the European Molecular Biology Organization (EMBO ALTF 241-2017). T.K. is supported by a Lise-Meitner fellowship from the Austrian Science Fund (FWF M2403). D.A. and Cs.B. are supported by the K119950, KH17-126718, NVKP_16-1-2016-0004, and NVKP_16-1-2016-0005 grants of the Hungarian National Research, Development and Innovation Office, the Janos Bolyai research scholarship, and the LP95021 grant of the Hungarian Academy of Sciences. C.S. was supported by a Feodor Lynen Fellowship of the Alexander von Humboldt Foundation. C.B. is supported by a New Frontiers Group award of the Austrian Academy of Sciences and by an ERC Starting Grant (European Union's Horizon 2020 research and innovation programme, grant agreement n° 679146).

## Author contributions

A.F.R., T.K., D.A., C.S. and Ch.B. designed the study; S.T., M.R., Z.M. and Cs.B. provided samples and clinical data; T.K., F.Z., T.P. and C.S. performed the experiments with contributions from M.F., L.C.S., A.N. and D.A.; A.F.R. and N.F. analyzed the data with contributions from T.K. and Ch.B.; Ch.B. supervised the research. A.F.R., T.K., N.F., Z.M., Cs.B., D.A., C.S. and Ch.B. wrote the manuscript with contributions from all authors.

## Competing interests

The authors declare no competing interests.
