## [Peer Review File · Nature Communications]

Reviewers' comments:

Reviewer #1 (Remarks to the Author):

In their updated manuscript, Rendeiro, Krausgruber and colleagues have refined their analysis of immunophenotypes, bulk chromatin accessibility, and single cell transcriptomes of CLL patients in response to ibrutinib treatment.

The authors have addressed several of the minor and a few of the major concerns from the previous round of review, with some additional analyses. For example, the DNA copy profile inference in the newly added Figure 1f, expanded upon with the analysis of subclonal genetic heterogeneity (Figure 4 and Supplementary Figure 10), extends the power of their previous analysis while continuing to use data already in hand.

They present sound statistical analyses of high quality data, with findings validated both internally from orthogonal data sets as well when compared with published data in more recent literature. However, concerns remain regarding the novelty and interpretation of their findings in the context of the current literature (though I do appreciate the fact that the authors rightfully point out several of the limitations in the present study).

Comments

The newly added analysis demonstrating the association between chromatin state at day 0 and response to ibrutinib at day 120 is enticing (Figure 4c-d and Supplementary Figure 11). For the genomic regions separating the fast and slow response, was there a clear relation to gene expression signatures? Given that the authors follow this section with gene expression signatures that capture the molecular response (Figure 4e-g), it would be interesting to know if these signatures can be placed in the context of the ATAC data at day 0, or vice versa.

Figure 3 starts with chromatin accessibility dynamics in non-malignant cells, then transitions to the common signature of the 165 genes, by gene expression. Given the enrichment of regulatory regions in Figure 3d, and the authors' own ATAC data, does the integration of chromatin data (via ChIP-seq or motif signatures) lend explanatory power to different (or the same) enhancer elements controlling the same gene expression changes in different cell types? Is there a corresponding ATAC dynamic related to these 165 genes?

As explained in the methods section for ATAC data processing, the authors observed "an association of several (but not all) samples from one processing batch with a specific principal component, while we did not observe any association of these samples with any known biological factor" (referring to now Supplementary Figure 7), after which they proceed to remove the latent variable (first principal component) before renormalization. Despite the lack of association with any known biological factor, how confident can we be that real biological signal is not being discarded in this process? Would surrogate variable analysis (SVA and ComBat in R) yield similar results?

In Figure 4a-b, the change in genetic diversity is measured as the change from day 0 to day 120 (specifically, 120 - 0), and the authors note a positive correlation between genetic diversity and cellular response to ibrutinib, with CLL5 as the standout case with the most genetic diversity and the largest reduction by flow cytometry. Was this the expected outcome? I may be misunderstanding the current literature, but I would have expected enhanced proliferation in the face of therapy (e.g. CLL6) to display increased genetic diversity, and responders (e.g. CLL5, presumably with the elimination of some subclonal populations) to show decreased diversity. Please correct me if I'm wrong on this point.

While I appreciate what appear to be consistently sound statistical approaches employed here,

there is still a reliance on z-score when demonstrating trends. As pointed out in the first round of reviews, getting a sense of the magnitude of baseline state and changes is important. Perhaps this can be demonstrated by presentation of several genome browser tracks of relevant genes/loci with a clearly demonstrated chromatin and expression dynamic, just to give the reader a sense of the magnitude of change.

As I commented on in the first round of review, motif signatures (via Homer) are mentioned in the methods section, but I didn't see the results referenced at all in the manuscript. While the LOLA analysis provided useful information with published ChIP-seq and other data sets, analysis of de novo or known motifs in the dynamic ATAC peaks could yield additional useful insight.

Reviewer #2 (Remarks to the Author):

The manuscript submitted by the group of C. Bock and collaborators deals with a novel strategy to map the temporal effect of cancer drugs on chromatin using single cell transcriptomics and chromatin profiling, and applies it to ibrutinib treatment in CLL as a model. I believe that the manuscript represents a significant advance in the field of CLL from 3 different perspectives, technological, biological and clinical, and furthermore shall have an impact in cancer therapeutics in general. I do not have any significant criticisms to make in this revised of the manuscript, which has clearly improved since I reviewed it for Nature Medicine.

Reviewer #3 (Remarks to the Author):

Thank-you for your thoughtful and detailed responses to my comments and questions. I still have a few remaining items to address, please.

1. Abstract last sentence (and other occurrences in the paper) I would avoid calling ibrutinib therapy "therapeutic B cell receptor inhibition". Not the BCR itself is the target, it is BTK, so you could say BCR signaling-, or BTK-inhibition.
2. Page 2, 2nd paragraph: Please cite recent randomized studies (Woyach JA N Engl J Med. 2018 Dec 27; 379(26):2517-2528, Moreno C Lancet Oncol. 2019 Jan; 20(1):43-56. Doi) that demonstrate superiority of ibrutinib over standard chemotherapy-based treatment in CLL, and rephrase. Ibrutinib is becoming standard of care for most patients with CLL

Point-to-point reply to reviewer's comments

Reviewer #1

In their updated manuscript, Rendeiro, Krausgruber and colleagues have refined their analysis of immunophenotypes, bulk chromatin accessibility, and single cell transcriptomes of CLL patients in response to ibrutinib treatment.

The authors have addressed several of the minor and a few of the major concerns from the previous round of review, with some additional analyses. For example, the DNA copy profile inference in the newly added Figure 1f, expanded upon with the analysis of subclonal genetic heterogeneity (Figure 4 and Supplementary Figure 10), extends the power of their previous analysis while continuing to use data already in hand.

They present sound statistical analyses of high quality data, with findings validated both internally from orthogonal data sets as well when compared with published data in more recent literature. However, concerns remain regarding the novelty and interpretation of their findings in the context of the current literature (though I do appreciate the fact that the authors rightfully point out several of the limitations in the present study).

We thank the reviewer for the careful review and additional comments, and for appreciating the hard work that went into the revised manuscript. In this new version of our manuscript, we have addressed the remaining comments with further bioinformatic analyses and text edits. While this study is not without limitations, we discuss its limitations openly in our manuscript, and we provide the raw and processed data as well as detailed documentation of our bioinformatic analyses, which makes it easy for others to build upon our results.

Comments

The newly added analysis demonstrating the association between chromatin state at day 0 and response to ibrutinib at day 120 is enticing (Figure 4c-d and Supplementary Figure 11). For the genomic regions separating the fast and slow response, was there a clear relation to gene expression signatures? Given that the authors follow this section with gene expression signatures that capture the molecular response (Figure 4e-g), it would be interesting to know if these signatures can be placed in the context of the ATAC data at day 0, or vice versa.

We thank the reviewer for the suggestion, which we are addressing by a new section and a new supplementary figure (**Supplementary Figure 13**) in our revised manuscript, as well as an extension of the Methods section.

In order to compare the signatures of molecular response discovered in the two data types (scRNA-seq and ATAC-seq), we analyzed in more detail the regression model that predicts the time since the start of treatment based on the scRNA-seq data. **Supplementary Figure 13a-c** illustrates how the regression coefficients (which quantify change of gene expression over time) can be used to investigate patients and genes over the whole course of treatment. Furthermore, by correlating the regression coefficients across patients with the observed individual response of the patients to ibrutinib, we were able to classify genes based on whether their change over time is indicative of slow or faster response to therapy (**Supplementary Figure 13d-e**).

We further investigated the relationship between the ATAC-seq signature discovered from day 0 samples and the regression analysis using scRNA-seq, assessing the overlap between the sets of genes in every signature using Fisher's exact test (**Figure R1**).

Figure R1: Relationship between the identified molecular signatures of ibrutinib response.

Heatmaps showing the pairwise overlap between the identified molecular signatures of ibrutinib response. The left panel displays the log2 odds ratio for the pairwise overlap between signatures, and the right panel displays the corresponding FDR-adjusted p-values based on Fisher’s exact test. “ATAC-seq day 0 signature” refers to the genes associated with the regulatory regions of the signature in Figure 4c/Supplementary Figure 11b. “scRNA-seq cross-cell type signature” refers to the signature in Figure 3e.

Our overlap analysis between the gene signatures (**Figure R1**) showed that the differential analysis of gene expression over time, the signature derived from differentially expressed genes across cell types (**Figure 3e**), and the regularized regression approaches show substantial overlap, while the day 0 ATAC-seq signature is clearly distinct from the other signatures. This observation supports that ATAC-seq indeed captures a different facet of the cell states associated with the response to ibrutinib than gene expression profiling.

We also compared the estimated regression coefficients per patient to our surrogate measure of the response to ibrutinib treatment – the amount of CLL cell reduction at day 120 of treatment. This way, we can assess whether the change (up-/down-regulation) of a gene over time is associated with a fast or slow response to ibrutinib (**Supplementary Figure 13**).

Figure R2: Signature discovered from ATAC-seq samples at day 0, with chromatin accessibility values aggregated per gene. The separation of patients according to the reduction of CLL cells at day 120 is still visible.

Finally, we now show that the ATAC-seq signature partially recapitulates the similarity between patients in their expression profiles at day 0 of treatment (**Figure R3**), although this analysis is limited in power given that we do not have scRNA-seq data for all patients for which we have ATAC-seq profiles at day 0.

Figure R3: Expression levels of genes associated with the signature discovered from ATAC-seq samples at day 0, based on the respective scRNA-seq samples at day 0. For visualization, a random sample of gene labels are displayed (x axis labels), and expression levels were averaged across cells for each sample.

Figure 3 starts with chromatin accessibility dynamics in non-malignant cells, then transitions to the common signature of the 165 genes, by gene expression. Given the enrichment of regulatory regions in Figure 3d, and the authors' own ATAC data, does the integration of chromatin data (via ChIP-seq or motif signatures) lend explanatory power to different (or the same) enhancer elements controlling the same gene expression changes in different cell types? Is there a corresponding ATAC dynamic related to these 165 genes?

Regarding the more general question of whether genes differentially regulated over time across cell types utilize different regulatory elements, we have previously observed considerable overlap in differentially regulated genes (**Figure 1h and Supplementary Figure 4**) and pathways (**Figure 1i and Supplementary Figure 6**) across cell types, we also found relatively low overlap in terms of the regulatory elements changing (**Supplementary Figure 9d**).

Specifically regarding the 165 genes of the cross-cell-type signature, we now display the chromatin accessibility levels of regulatory elements associated with those genes (**Figure R4**). In addition to cell-type specific differences, which dominate the analysis, we also observed dynamic changes upon treatment (e.g. in CLL cells and CD8 T cells, as shown in the last two heatmaps of **Figure R4**). The correlation between the single-cell RNA-seq and the ATAC-seq data was far from perfect, consistent with our previous observations, which we discuss in the manuscript and in our previous response to comments from Reviewer 3. However, we would like to emphasize this 165 gene cross-cell-type signature of ibrutinib response was successfully validated in an independent cohort of larger size (Figure 3g, h), adding further confidence to this result and signature.

Figure R4: Changes in chromatin accessibility for differentially expressed genes. Heatmap of gene-level chromatin accessibility for the genes in the cross-cell-type signature shown in Figure 3e. Chromatin accessibility represents the mean of the regulatory elements associated with a gene. The first plot shows cross-cell-type gene-level intensities, while the two other show CD8⁺ T cells and CLL cells, respectively.

As explained in the methods section for ATAC data processing, the authors observed “an association of several (but not all) samples from one processing batch with a specific principal component, while we did not observe any association of these samples with any known biological factor” (referring to now Supplementary Figure 7), after which they proceeded to remove the latent variable (first principal component) before renormalization. Despite the lack of association with any known biological factor, how confident can we be that real biological signal is not being discarded in this process? Would surrogate variable analysis (SVA and ComBat in R) yield similar results?

Several lines of evidence made us confident that that this was indeed a technical effect, and that we addressed it adequately with the type of correction used in our manuscript. Nevertheless, we agree with the reviewer that SVA and ComBat analysis could provide another level of validation. We have now performed the analysis suggested by the reviewer and compared its results in detail with our approach (as described below). This analysis provides further support for the reliability of our approach. We thank the reviewer for the suggestion!

The association with processing batch in the uncorrected ATAC-seq data is illustrated in **Figure R5**, where there is a group of samples (highlighted in the figure) that belong to several cell types but nevertheless separate consistently from the remaining samples along the principal component 3 (PC3) axis. These samples were part of two experimental processing batches (“ATACTK078” and “ATACTK070”). Since this group of samples did not associate with any known annotated biological or technical variable and since it was seen across several cell types and patient samples, we decided to treat it as a latent variable of likely technical origin.

Figure R5: Principal component analysis of the ATAC-seq data prior to batch effect correction. Each combination of successive principal components is plotted, with samples colored according to known variables. Colored circles represent samples, and squares represent the centroids of each group. The samples highlighted with a circle represent a group of samples affected by a latent factor of likely technical origin.

Our approach to correct the ATAC-seq data for this latent variable is based on the fact that it was well captured by PCA, suggestion that it should be possible to remove the latent vector using straightforward matrix operations. To account for cell type-specificity, we regressed out the latent vector in each cell type separately (excluding monocytes given that no sample was affected) as described in the Method section. The results of the correction on the batch effect correction using matrix operations is shown in **Figure R6, left side**. We color-coded samples affected by the latent factor prior to the correction in the “latent_variable” column of the plots.

We then compared our approach (as described above) with the SVA/ComBat approach suggested by the reviewer. ComBat works by removing the influence of a known, user-specified factor from the data. To make sure that the removal preserves as much of the variables of interest (e.g. cell type-, patient-, and timepoint-specific variability), it is possible to supply ComBat with a model matrix that includes these variables. In our case however, including a model matrix with all parameters of interest (patient ID, sex, timepoint, cell type, IGHV mutation, speed of response to ibrutinib, among others) was not possible because not enough independent combinations of these dimensions are represented in our dataset. We therefore prompted ComBat to correct the latent factor discovered in the PCA without any further customization (**Figure R6, right side**). ComBat corrected successfully for the latent variable, with very similar outcomes as our own method (**Figure R6**).

Figure R6: Principal component analysis of the ATAC-seq data after batch effect correction. Two independent methods were employed: A custom method based on matrix operations and the published ComBat tool. Each combination of successive principal components is plotted, with samples colored according to known variables or the “latent_variable”, which is derived from samples that separated along PC3 of **Figure R5**. Note that some principal components seem inverted between the two methods (e.g. PC1, PC5).

For a more quantitative comparison of the two approaches to batch effect correction, we computed the association with known technical and biological factors using pairwise Kruskal-Wallis tests for each factor’s level. **Figure R7** displays smallest FDR adjusted p-values for each possible factor / principal component association.

Figure R7: Association of principal components with biological and technical factors in the analysis. The green diamonds mark associations which are significant at $p < 0.01$ with FDR-based adjustment.

Several observations can be made from **Figure R7**: (i) the most consistent and some of the strongest associations are with cell type; (ii) the association between principal component 3 and the latent factor is reduced by both methods; (iii) ComBat completely removed any trace of association of the latent factor with the principal components, while the PCA-based method was milder on this aspect; (iv) small differences are observed for the association of patient-specific variation with other principal components, which are at least in part due to the less aggressive correction for the latent variable using the PCA-based method.

In summary, both our PCA-based method and ComBat yielded similar results. Given that the PCA-based method is milder and more tailored to the dataset, we decided to continue to focus on the PCA-based method for batch effect correction in our study, while the ComBat analysis provided a valuable external validation.

In Figure 4a-b, the change in genetic diversity is measured as the change from day 0 to day 120 (specifically, $120 - 0$), and the authors note a positive correlation between genetic diversity and cellular response to ibrutinib, with CLL5 as the standout case with the most genetic diversity and the largest reduction by flow cytometry. Was this the expected outcome? I may be misunderstanding the current literature, but I would have expected enhanced proliferation in the face of therapy (e.g. CLL6) to display increased genetic diversity, and responders (e.g. CLL5, presumably with the elimination of some subclonal populations) to show decreased diversity. Please correct me if I'm wrong on this point.

It is true that high genetic diversity is often considered a risk factor for therapy resistance, across a broad range of cancers. However, for the combination of CLL (a slowly evolving cancer in which rapid evolutionary sweeps are uncommon or absent) and ibrutinib (a drug that has shown similar efficacy across CLL patient subgroups defined by different genetic aberrations and complexity), it is less clear how the association between genetic diversity and the response to therapy might play out. We therefore did not have any strong assumptions about the expected outcome when we performed this analysis.

The results of our analysis indeed suggest that an increase in genetic diversity between day 0 and day 120 was positively correlated with the reduction of CLL cells at day 120/150 (**Figure 4a-b**). We have carefully considered potential explanations for this observation, and the most plausible explanation appears to be the following: For those cases with the most dramatic reduction in CLL cells at day 120/150, ibrutinib seems to work particularly well against the dominant CLL clone; and as the result of the rapid depletion of the dominant CLL clone, minority clones that contribute little to the CLL cell profile at day 0 become clearly detectable at day 120, thus causing an observed increase in genetic diversity between day 0 and day 120.

While I appreciate what appear to be consistently sound statistical approaches employed here, there is still a reliance on z-score when demonstrating trends. As pointed out in the first round of reviews, getting a sense of the magnitude of baseline state and changes is important. Perhaps this can be demonstrated by presentation of several genome browser tracks of relevant genes/loci with a clearly demonstrated chromatin and expression dynamic, just to give the reader a sense of the magnitude of change.

We fully agree with the reviewer that it is important to report both statistical significance (e.g., as FDR-corrected p-value or z-score) and effect size (e.g., as odds ratio or absolute differences), and we provide such measures throughout our manuscript. We also agree that genome browser tracks of individual genes/loci can be useful to provide a visual intuition of effect size (albeit often at the cost of some cherry-picking). However, our dataset's batch effect / latent variable issue implies that the raw genome browser tracks are not directly interpretable. We are not aware of a statistically sound and computationally feasible way of batch effect correction of entire genome browser tracks that would ensure straightforward quantitative and reliable interpretability. We therefore decided to not use genome browser tracks anywhere in our manuscript and instead focus on the batch effect corrected data that provide the basis for our figures and supplementary tables.

As I commented on in the first round of review, motif signatures (via Homer) are mentioned in the methods section, but I didn't see the results referenced at all in the manuscript. While the LOLA analysis provided useful information with published ChIP-seq and other data sets, analysis of de novo or known motifs in the dynamic ATAC peaks could yield additional useful insight.

We apologize for the confusion. In the previous round of manuscript revisions, we performed *de novo* motif enrichment analyses using HOMER as suggested by Reviewer 3. This analysis found good agreement between the region overlap analysis (based on LOLA) and the *de novo* enrichment analysis (based on HOMER). Given the length of the manuscript and the partial redundancy of the HOMER results, we included these data only in the previous Response to the Reviewers, but not in the main manuscript. However, we forgot to remove the mentioning of HOMER from the Methods section, which gave rise to the current confusion. We have now corrected this oversight, and we thank the reviewer for the careful reading and helpful comments!

Reviewer #2:

The manuscript submitted by the group of C. Bock and collaborators deals with a novel strategy to map the temporal effect of cancer drugs on chromatin using single cell transcriptomics and chromatin profiling, and applies it to ibrutinib treatment in CLL as a model. I believe that the manuscript represents a significant advance in the field of CLL from 3 different perspectives, technological, biological and clinical, and furthermore shall have an impact in cancer therapeutics in general. I do not have any significant criticisms to make in this revised of the manuscript, which has clearly improved since I reviewed it for Nature Medicine.

We thank the reviewer for the positive comments.

Reviewer #3

Thank you for your thoughtful and detailed responses to my comments and questions.

I still have a few remaining items to address, please.

1. Abstract last sentence (and other occurrences in the paper) I would avoid calling ibrutinib therapy "therapeutic B cell receptor inhibition". Not the BCR itself is the target, it is BTK, so you could say BCR signaling, or BTK-inhibition.
2. Page 2, 2nd paragraph: Please cite recent randomized studies (Woyach JA N Engl J Med. 2018 Dec 27;379(26):2517-2528, Moreno C Lancet Oncol. 2019 Jan;20(1):43-56. Doi) that demonstrate superiority of ibrutinib over standard chemotherapy-based treatment in CLL, and rephrase. Ibrutinib is becoming standard of care for most patients with CLL

We thank the reviewer for appreciating our detailed responses and for the final comments, which we have addressed in the final version of our manuscript.

REVIEWERS' COMMENTS:

Reviewer #1 (Remarks to the Author):

I thank the authors for the thoughtful consideration of my comments and questions. The follow up explanations and analyses addressed all of my concerns. I believe the manuscript represents both a significant advance for CLL research as well as a valuable resource for the CLL community and beyond.